# Integrating mass spectrometry with MD simulations reveals the role of lipids in Na$^+$/H$^+$ antiporters

Michael Landreh[1], Erik G. Marklund[1,2,*], Povilas Uzdavinys[3,*], Matteo T. Degiacomi[1], Mathieu Coincon[3], Joseph Gault[1], Kallol Gupta[1], Idlir Liko[1], Justin L.P. Benesch[1], David Drew[3] & Carol V. Robinson[1]

Na$^+$/H$^+$ antiporters are found in all kingdoms of life and exhibit catalysis rates that are among the fastest of all known secondary-active transporters. Here we combine ion mobility mass spectrometry and molecular dynamics simulations to study the conformational stability and lipid-binding properties of the Na$^+$/H$^+$ exchanger NapA from *Thermus thermophilus* and compare this to the prototypical antiporter NhaA from *Escherichia coli* and the human homologue NHA2. We find that NapA and NHA2, but not NhaA, form stable dimers and do not selectively retain membrane lipids. By comparing wild-type NapA with engineered variants, we show that the unfolding of the protein in the gas phase involves the disruption of inter-domain contacts. Lipids around the domain interface protect the native fold in the gas phase by mediating contacts between the mobile protein segments. We speculate that elevator-type antiporters such as NapA, and likely NHA2, use a subset of annular lipids as structural support to facilitate large-scale conformational changes within the membrane.

[1] Department of Chemistry, Physical & Theoretical Chemistry Laboratory, University of Oxford, South Parks Road, Oxford, Oxfordshire OX1 3QZ, UK. [2] Department of Chemistry–BMC, Uppsala University, Box 576, Uppsala SE-751 23, Sweden. [3] Centre for Biomembrane Research, Department of Biochemistry and Biophysics, Stockholm University, Stockholm SE-106 91, Sweden. * These authors contributed equally to this work. Correspondence and requests for materials should be addressed to D.D. (email: ddrew@dbb.su.se) or to C.V.R. (email: Carol.Robinson@chem.ox.ac.uk).

Na$^+$/H$^+$ antiporters regulate intracellular pH, cell volume and sodium content by rapidly exchanging Na$^+$ or Li$^+$ in exchange for protons across the membrane[1]. Their dysfunction is associated with a number of diseases, and they are well-established drug targets[1,2]. On the basis of the prototypical Na$^+$/H$^+$ antiporter structure of E. coli NhaA, it was established that Na$^+$/H$^+$ antiporters are made up of a core ion-translocation domain and a scaffold dimer domain[3–5]. More recently, crystal structures of another antiporter (NapA from T. thermophilus) have been determined in both outward- and inward-facing conformations. On the basis of these crystal structures, it was shown that a rigid-body movement of the core domain, relative to the dimeric scaffold, displaces the substrate-binding site some 10 Å across the membrane in an elevator-like manner[6,7]. During these structural transitions, the scaffold domain is thought to stay anchored in the membrane, whereas the core domain is freely able to move as it is only loosely attached via flexible hinges to the dimer domain. Such elevator-like structural transitions against a fixed (hydrophobic) dimer domain surface imply that lipids might anchor the dimer scaffold, while facilitating movement of the core domain. Similar structural transitions have also been observed for other unrelated transporters, such as the sodium-coupled glutamate symporter homologue Glt$_{Ph}$ (refs 2,8), but the role of protein–lipid interactions in facilitating these types of movements are yet to be understood.

Mass spectrometry (MS) enables us to study the effects of individual lipids on the structures and stabilities of detergent-solubilized membrane proteins[9,10]. In short, following transfer into the gas phase, intact membrane proteins are released from protective detergent micelles by collisional activation, and individual oligomeric states and lipid-bound populations can be identified based on their mass-to-charge ($m/z$) ratio[11]. Here we compare stabilities and endogenous lipid binding of homologous Na$^+$/H$^+$ antiporters, and reveal similarities between NapA and human NHA2, but distinct differences to E. coli NhaA. Monitoring collisional unfolding by (ion mobility–MS) IM–MS informs about the conformational stability of the individual protein complex populations[12–14]. By employing IM–MS, we show that lipids confer resistance and restrict unfolding of NapA. MD simulations suggest that these lipids occupy a subset of annular positions, where they provide contacts between the core and dimer domains and stabilize the flexible domain connections. We conclude that lipids specifically pack around the mobile segments of elevator-type antiporters and provide structural support that facilitates the large-scale conformational changes during transport.

## Results

**MS shows differences between human and bacterial antiporters.** Purification and functional reconstitution of bacterial NapA and NhaA into liposomes have been well documented[6,15]. Previously, mammalian Na$^+$/H$^+$ exchangers could not be functionally reconstituted owing to difficulties in isolation of suitable material. Recently, we have been able to purify human NHA2 and recover robust transport activity in liposomes (Uzdavinys et al., manuscript submitted). We therefore subjected NHA2 and the model systems NapA and NhaA to MS analysis. To be able to investigate the structure and interactions of a group of antiporters, we first screened for conditions that maintain intact complexes of all three proteins in the gas phase. Even the most detergent-stable NapA dimer dissociate readily, when the protein is released from n-dodecyl β-D-maltopyranoside (DDM) micelles through collisional activation

in the ion trap of the mass spectrometer. The polyethylene glycol (PEG)-detergent C$_{12}$E$_9$, on the other hand, enabled us to observe intact dimers of all three antiporters (Fig. 1).

NHA2 gave rise to a mass spectrum that assigned predominantly to a dimeric protein (Fig. 1a). The low mass spectral resolution is consistent with multiple salt, detergent and potentially lipid adducts. Attempts to improve spectral quality through harsher MS conditions resulted in loss of the MS signal, which is attributable to the sensitivity common among detergent-solubilized mammalian membrane proteins. Residual NHA2 monomers do not retain a significant amount of bound lipids, as only a minor adduct peak with a mass consistent with a phospholipid binding was observed (Fig. 1b). In line with this, the dominant species is a dimer that corresponds in mass to that of two lipid-free monomers (Supplementary Table 1). The NHA2 dimer remains largely intact at increasing activation energy, which implies that the protein has a stable dimer interface (Supplementary Fig. 1).

Having established the mass spectral properties of NHA2, we turned our attention to the bacterial NapA antiporter. As for NHA2, mass spectrum of NapA shows two distinct features. First, the protein forms dimers that remain stable even at high collision voltages (Fig. 1c; Supplementary Fig. 1), consistent with the high stability of the antiporter from a thermophilic organism. Second, NapA monomers bind only very small amounts of lipids with an average mass corresponding to the most common phospholipids found in E. coli, and the mass of the dimer corresponds to that of two lipid-free monomers (Fig. 1d; Supplementary Table 1).

In contrast, the mass spectra recorded for E. coli NhaA revealed pronounced differences in dimer stability and lipid binding (Fig. 1e). Even under the softest ionization conditions, the majority of the protein was found to be monomeric, and only a minor population was released from the detergent as intact dimers. Increasing the collision voltage completely dissociated the remaining dimers (Supplementary Fig. 1). These observations correlate well with the weak dimer interface in the crystal structure, which is made up interactions between short β-hairpins on the extracellular side with few other contacts between monomers[16]. We were able to resolve a co-purified lipid bound to monomeric NhaA, which based on the mass of 1,430 Da and the spacing of 28 Da between acyl-chain lengths could be identified as cardiolipin (CDL; Fig. 1f). Interestingly, comparing this data with the characterization of a total lipid extract from E. coli reveals that the most predominant CDL species in the E. coli membrane has a composition of 66:2 (average mass 1,376.8 Da)[17]. The predominant CDL bound to NhaA has a mass of 1,430 Da, corresponding to CDL 70:3 (average mass 1,430.9 Da). The long chain length indicates that NhaA recruits lipids with long side chains with respect to the average CDL.

Together, our MS data show that NapA and NHA2 share a stable dimer architecture that is largely independent of lipid interactions, while NhaA exhibits differences in lipid preferences and dimer stability.

**Covalent links between domains restrict unfolding of NapA.** Having established that the mass spectrum of the elevator-type antiporter NapA (Fig. 2a) resembles human NHA2, we subjected NapA to IM–MS analysis. The collision cross-section (CCS) values measured by IM–MS are 4,550 and 4,580 Å$^2$ for the 11$^+$ and 12$^+$ charge states, respectively[18]. These values indicate some compaction compared with the value of 4,990 Å$^2$ calculated from the crystal structure. Such collapse is well documented in IM–MS and is often associated with the

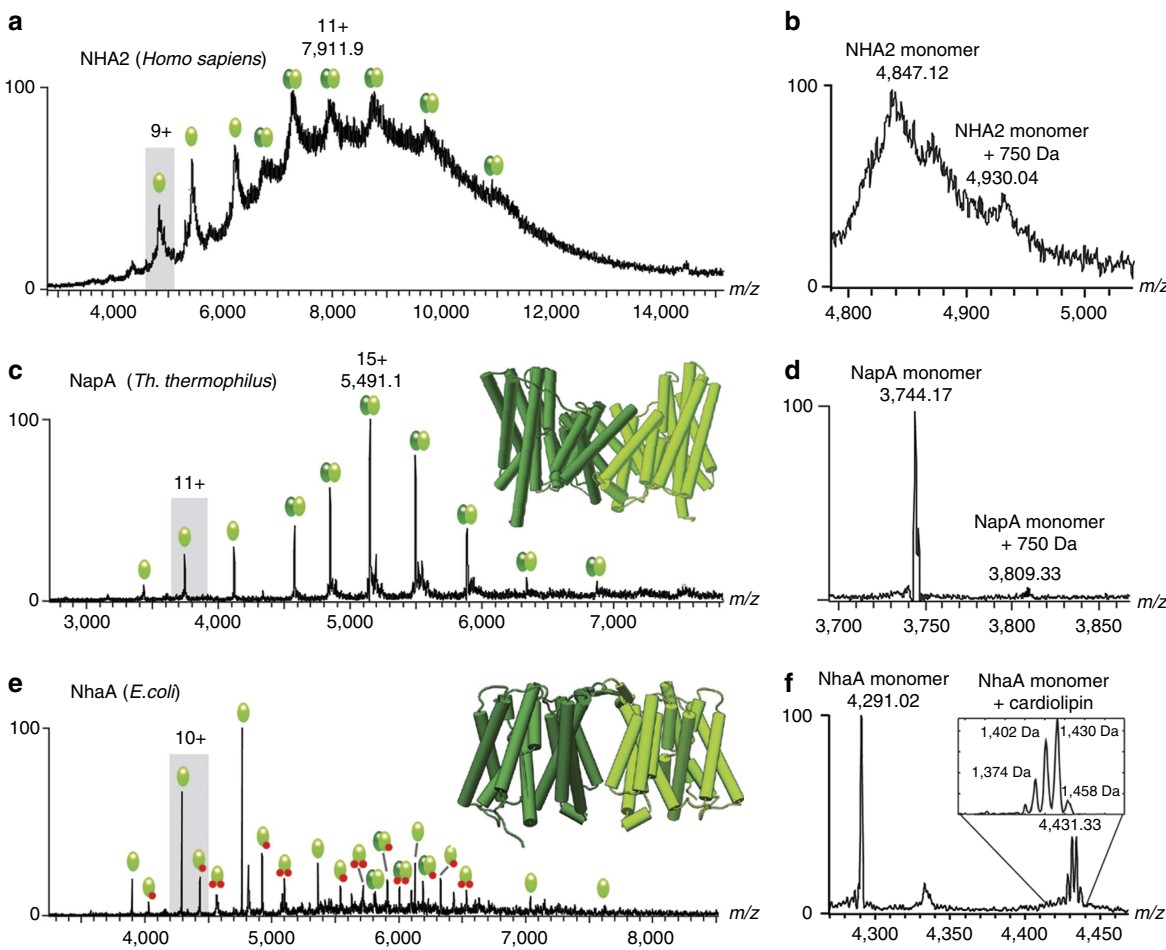

**Figure 1 | MS of homologous sodium/proton antiporters in $C_{12}E_9$ under gentle ionization conditions reveals similarities between NHA2 and NapA.**
(**a**) NHA2 appears largely as intact dimers after detergent release. The measured mass of the main dimer species corresponds to that of two lipid-free monomers. Green spheres indicate proteins, and red spheres bound lipid molecules. (**b**) NHA2 retains only minor lipid adducts after delipidation, indicating low-binding selectivity. (**c**) Like NHA2, NapA appears as an intact dimer following release from detergent micelles, in agreement with the crystal structure (insert). (**d**) NapA does not retain a notable amount of co-purified lipids. (**e**) NhaA appears almost exclusively as monomer following detergent removal, in line with the sparse subunit contacts in the crystal structure (insert). (**f**) Lipid adducts retained by NhaA monomers could be identified as CDL in deconvoluted spectra (insert). All spectra were recorded at a collision voltage of 50 V.

$\sim 10\%$ decrease in CCS observed in MD simulations (Supplementary Table 2) and is attributed to the collapse of surface residues and unsupported loops in the gas phase[19]. The consistency between calculated and experimental values indicates NapA largely retains its native conformation and is therefore not unfolded in PEG detergents during transition into the gas phase.

We subsequently used collision-induced unfolding to investigate the conformational stability of NapA in the gas phase. For wild-type NapA, increasing the activation energy from 700 to 1,500 eV in the collision cell and monitoring the CCS via IM–MS revealed an increase consistent with unfolding (Fig. 2b). This unfolded form has a maximum CCS of 5,730 $\text{Å}^2$ for the $11^+$ charged state, which represents an increase of 21% over the compact state (Fig. 2c). We observed that unfolding is effectively a two-state process, with only a very minor population of intermediates between the compact native-like conformation and the unfolded conformation(s).

Recently, we were able to trap the cysteine-less wild-type NapA protein in predominantly a single conformation by introducing cysteine pairs between the core and dimer that spontaneously form a disulphide bond in a membrane[7].

Importantly, on reduction of the disulphide bond with dithiothreitol (DTT) it was found that NapA activity is recoverable, indicating that the formation of the disulphide does not overly perturb the integrity of the protein[7]. Interestingly, the unfolding trajectory of the outward-facing disulphide-trapped NapA protein (V71C/L141C) showed only a small shift to an unfolded conformation, being $\sim 12\%$ larger than the compact state (Fig. 2c). Furthermore, rather than unfolding in a single well-defined step, V71C/L141C showed a gradual increase in CCS, consistent with unfolding confined to a continuum of expanding states.

We next analysed a NapA variant where the core domains are able to move, but the dimer domains have now been covalently linked together by the introduction of a disulphide in isoleucine position 55. For the I55C variant that has wild-type-like activity[7], the CCS also increased during unfolding by $\sim 12\%$ over the native-like state. Indeed, the unfolding profile of I55C follows a similar trajectory to the V71C/L141C variant. Reduction of the disulphide in I55C with DTT results in an unfolding profile similar to that of wild-type NapA, in which the compact state is lost in a single step and the extended form reaches a higher CCS. The

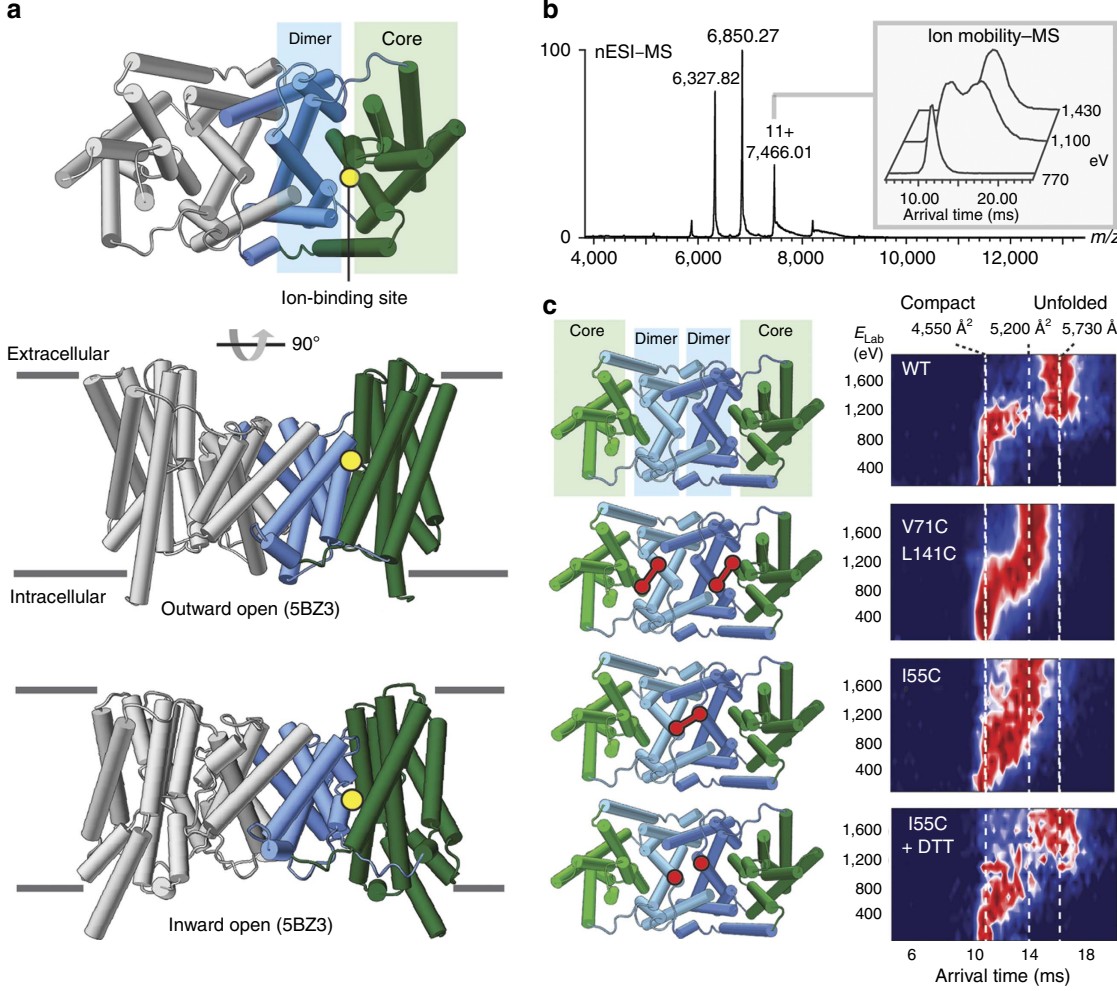

**Figure 2 | The unfolding trajectories of NapA variants show the effects of local stabilization.** (**a**) Structures of NapA in inward and outward-open states show the elevator-like movement of the ion-binding site (yellow). (**b**) Nano-electrospray ionization (nESI)–MS spectrum of dimeric NapA (molecular weight, 82,450 Da) in the detergent $C_8E_4$. Insert: ion mobility–MS shows unfolding in response to increasing activation energy. (**c**) Structures and unfolding of the NapA variants. The locations of the disulphide bonds that lock domain contacts are marked in red. Unfolding trajectories of the 11 + charge state of NapA with covalent bonds between the core and dimer domains (V71C/L141C) or dimer subunits (I55C) show a gradual unfolding transition and trapping of the unfolded proteins in states with reduced CCS compared with the wild type. The reduction of the disulphide bond abolishes both effects in the I55C mutant.

fact that the unfolding profile does not revert completely to NapA wild type can be attributed to the incomplete reduction of the disulphide bond by DTT, as shown previously[7]. Nevertheless, we can conclude that covalently linking either the dimer–dimer domains or the dimer–core domains restricts the extent of NapA unfolding by trapping the protein in intermediate-like unfolding states.

**Lipids stabilize compact states and restrict NapA unfolding.** Unusually the MS showed that virtually no endogenous lipids in NapA are co-purified after its extraction from *E. coli* membranes using the detergent DDM, which is contrary to the observations made for many other membrane proteins[20]. We, therefore, tested the lipid-binding ability of NapA by adding increasing amounts of *E. coli* lipids 1-palmitoyl-2-oleoyl-glycerophospho-ethanolamine (PE) or 1-palmitoyl-2-oleoyl-phosphatidyl-glycerol (PG), that have both been shown previously to preserve NapA function *in vitro*[6,7]. As shown in Fig. 3a, we observed a concentration-dependent increase of lipid protein complexes with no apparent saturation. Such binding behaviour is indicative of non-specific

phospholipid binding. We then tested the influence of lipid head-group charge through tandem MS of the protein–lipid complexes, that is, comparing the amount of energy required to dissociate the NapA–lipid interactions informs about the relative binding strengths in different protein–lipid complexes. Interactions with negatively charged PG lipids exhibit higher gas-phase stabilities than with zwitterionic PE, while uncharged monoacyl glycerol (9.9 MAG), that lacks the phosphate head group, readily dissociated with the detergent micelle (Fig. 3b,c; Supplementary Fig. 2). Thus, in the absence of charge, lipids are lost readily while negatively charged head groups promote lipid binding with higher gas-phase stability.

We further subjected lipid-free and lipid-bound NapA to collision-induced unfolding (Fig. 3d,e; Supplementary Fig. 3). We found that the lipid-bound protein unfolds to yield a broader arrival time distribution with an average CCS of $5,530 \text{ Å}^2$. This increase corresponds to an expansion of ~16% over the wild type. We also considered the resistance to unfolding of compact NapA populations conferred by binding to one PE or one PG lipid molecule. For the lipid-free protein, the compact population is lost at 900 eV. In contrast, the

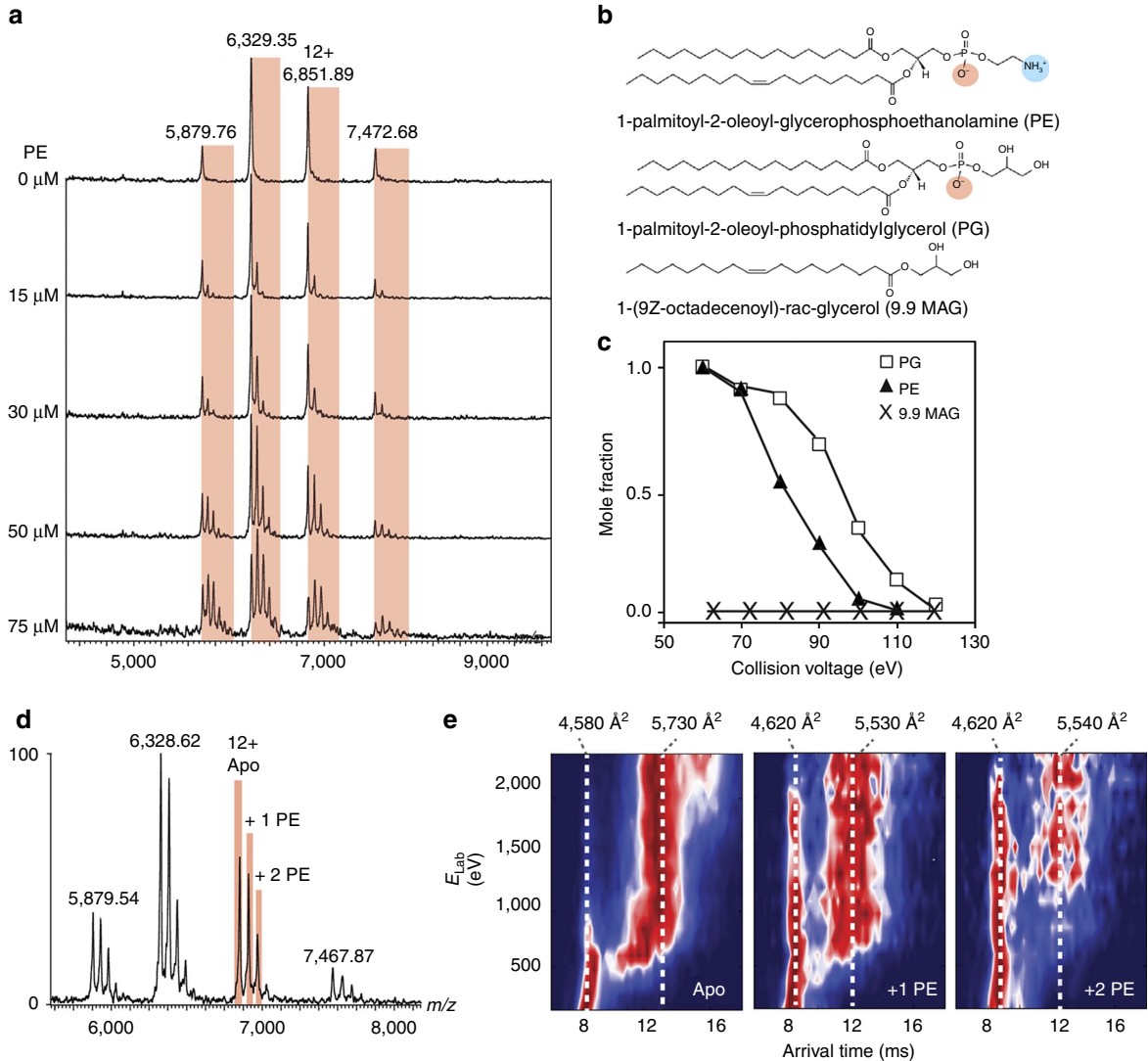

**Figure 3 | Lipids interact with NapA via charged head groups and restrict unfolding.** (**a**) nESI–MS spectra of NapA with PE show a concentration-dependent increase in the number of lipid adducts on the intact dimer (highlighted in red). (**b**) Structures of PG, PE and 9.9 MAG with positive and negative head-group charges indicated. (**c**) The dissociation curves from MS/MS reveal higher gas-phase stability of negatively charged PG that zwitterionic PE. The uncharged 9.9 MAG readily dissociates with the detergent. (**d,e**) Unfolding trajectories of the 12 + charge state of lipid-free and PE-bound NapA show that the protein–lipid complexes exhibit broader unfolding transitions and that the native-like population is lost at higher collision energies than the lipid-free form. Dashed lines indicate the CCS at 20 and 130 V, respectively.

compact lipid-bound populations can be detected up to 2,000 eV. The unfolding transition is extended significantly with both the compact and unfolded states coexisting across a broad energy interval. Similar observations were made for all charge states when PE or PG is bound to NapA (Supplementary Fig. 3).

Binding of a second PE or PG lipid to NapA shifts the onset of unfolding to yet higher energies > 2,300 eV. We interpret these results as showing lipid binding acts as a potent stabilizer of the compact NapA structure and that binding of the second lipid leads to a greater chance of lipid stabilization. Interestingly, lipid binding does not have a pronounced effect on the unfolding trajectories of the covalently linked NapA variants (Supplementary Fig. 4). We reason that these locked states are unable to unfold in a way that can be restricted by lipids, and the effects of covalent linkage and lipid binding are therefore not additive (Supplementary Fig. 4).

**Lipid interactions stabilize domain contact in the gas phase.** To identify potential lipid interaction sites, we performed 100-ns all-atom MD simulations of NapA in a PE bilayer. Over the course of the simulations, we found that the lipids do not distribute evenly around the protein, but exhibit a notable preference to cluster around the dimer domains and the flexible dimer–core domain connections composed of TM 6 and TM 9–10, leaving a relatively low density around the core domains (Fig. 4a). Due to the strong hydrophobic mismatch in this region, the membrane is highly compressed around the shorter dimer domains (Fig. 4b), reducing the thickness of the hydrophobic barrier that the ions have to traverse[2,7]. As expected, we observed in MD simulations that the lipids in these areas interact with membrane-adjacent positively charged residues located on the dimer domains and the hinge regions between dimer and core domains (Supplementary Fig. 5). The simulations therefore provide a rationale for the importance of head-group charge defined by MS (Fig. 3). Taken together,

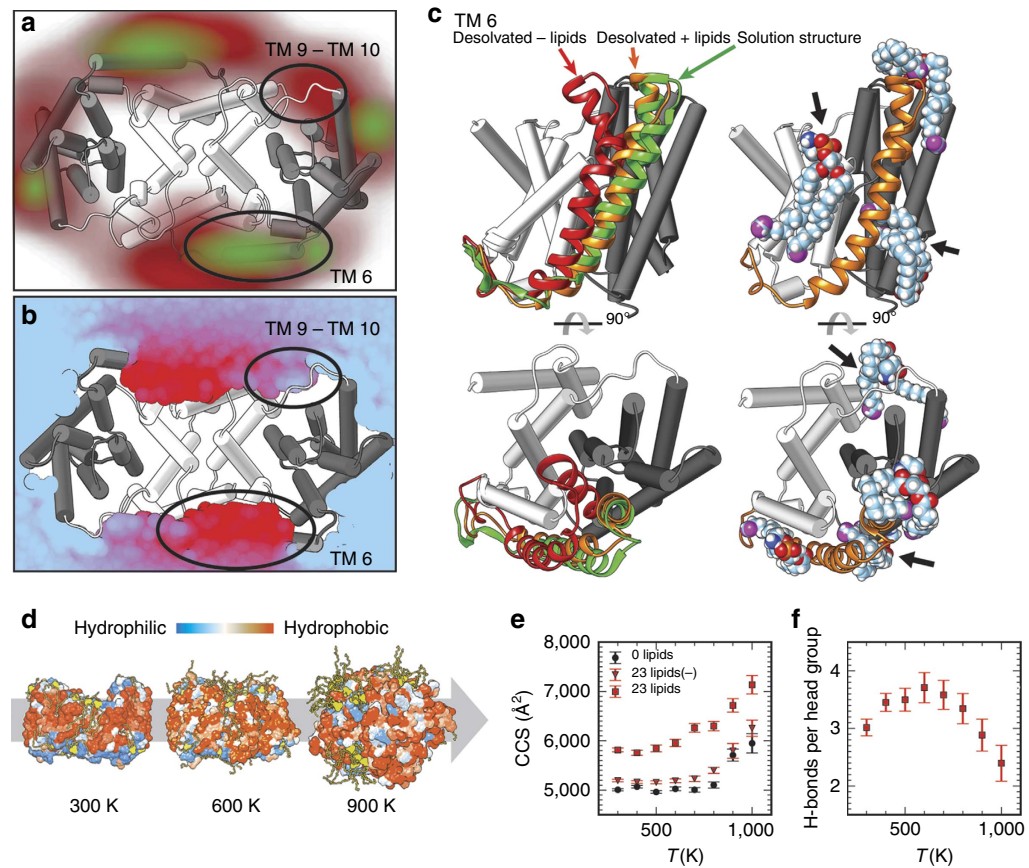

**Figure 4 | Lipids mediate dimer–core domain contacts and stabilize NapA in the gas phase.** (**a**) MD simulations of NapA in a PE bilayer show high phosphate densities around the dimer and core domain connections (TM 9–10, and TM 6). (**b**) The region around the core–dimer interface exhibits the highest degree of membrane compression. Densities on the inner and outer leaflets are coloured red and green, respectively. (**c**) Lipid binding stabilizes the domain contacts in the gas phase. In lipid-free NapA, TM 6, which connects core and dimer domains, unfolds rapidly (red). In the lipid-bound system (orange), lipids prevent structural collapse via head-group interactions and by intercalating into the interface between the core and dimer domains (black arrows). Core and dimer domains are shown in black and white, respectively. (**d**) Structural evolution of lipid-bound NapA at increasing temperatures shows a progressive loss of surface organization and the detachment of lipid acyl chains. Hydrophilic and hydrophobic surface areas are rendered in blue and orange, respectively, and lipids in yellow. (**e**) Effect of gas-phase unfolding of NapA with and without lipids on the theoretical CCS, based on the last 5 ns of each 10-ns simulation. The lipid-free system (0 lipids) maintains a native-like CCS up to a sudden ~20% increase at 800 K. The lipid-bound system (23 lipids) has a higher CCS due to the presence of lipid adducts. When the contribution from the lipids is not considered (23 lipids( − )), the CCS remains above that of the lipid-free protein, confirming that lipids reduce the structural collapse. (**f**) Before unfolding, each lipid head group forms on average three to four H-bonds with the protein. Error bars in **e**,**f** represent the s.d.'s over the last half of each simulation.

MD reveals that NapA preferentially interacts with lipids that tightly pack into the connection between dimer and core domains.

We speculated that the most preferred lipid interactions observed by MD stabilize the protein in MS. We therefore carried out *in vacuo* MD simulations of detergent-free NapA with and without PE bound to mimic conditions inside the mass spectrometer[21,22]. We therefore selected 23 PE molecules to represent the lipids in the first annular belt as observed in MD (Methods). The simulations enabled us to screen multiple protein–lipid interactions concurrently, while giving individual lipids sufficient space to rearrange on the surface of the protein in response to the gas-phase conditions. We then compared the NapA structures from the PE bilayer simulation before and after 20 ns in the gas phase. While the overall structure was found to remain largely unaffected, the flexible hinges between core and dimer domain, where the highest phosphate head-group density is observed, readily collapse in the gas phase. Lipid binding to TM 6 and between TM 9 and 10, on the other hand, significantly reduces unfolding of the desolvated protein

by intercalating between core and dimer domains and interacting with near-by charged residues (Fig. 4c; Supplementary Fig. 6).

Next, we performed multiple simulations of NapA *in vacuo* at different temperatures to obtain qualitative insights into the thermal unfolding process (Fig. 4d)[21–24]. CCS calculations of the equilibrated protein systems at the different temperatures allowed us to compare the predicted events with the unfolding steps observed by IM–MS. We find that NapA largely retains its compact conformation at temperatures < 800 K. At higher temperatures, the CCS suddenly increases by ~20%, analogous to the IM–MS unfolding trajectory of lipid-free NapA (Fig. 4e), although the simulations at high temperatures provide only a rough approximation for the unfolded structures. The stability of the CCS for the lipid-coated protein with respect to temperature up to 800 K suggests that no substantial rearrangements with respect to the overall lipid structure occur on intermediate timescales.

Next, we included the annular lipids in the simulation. At low temperatures, all lipids remain fully attached to the protein. The compact state of the lipid-stabilized system shows a

consistently higher CCS when the protein without the contribution from the bound lipid molecules themselves is considered, in line with the suggestion that lipids stabilize the overall conformation of the protein by supporting the collapse of unsupported segments. With increasing temperature, the acyl chains of the lipids detach from the transmembrane region. The head groups, on the other hand, remain attached to the same binding sites via an average of three to four hydrogen bonds to the protein that become perturbed only at the highest temperatures (Fig. 4f). These dynamics further support that our simulations capture the important lipid rearrangements on at least intermediate timescales. We conclude that the hydrogen bonding networks formed by the lipid head groups reduce the build-up of destabilizing structural rearrangements that occur during collisional activation. Together these results imply that lipids at the core–dimer interface prevent the collapse of flexible regions after transfer to the gas phase, while charge interactions between the phosphate head groups and basic residues surrounding the dimer domains enhance stability during thermal unfolding by providing additional molecular contacts.

## Discussion

In this study, we have used IM–MS and MD simulations to characterize the stability and lipid binding of the $Na^+/H^+$ antiporter NapA in relation to its human and bacterial homologues. NapA is arguably the structurally best-characterized antiporter, with structures determined in both major conformations at a functionally relevant pH, showing that large conformational changes of the core domains are required to mediate ion exchange[6,7]. We found that engineered covalent links between core and dimer domain restricts unfolding of NapA. Notably, the introduction of a disulphide bond in of itself does not restrict gas-phase unfolding in other protein systems[25], implying that the covalent linkages used here are influencing specific unfolding events in NapA. These restricted unfolding results imply that the core domains in the native protein are almost certainly flexibly attached, and these interactions are disrupted during gas-phase activation.

By using MS to inform MD simulations, we found that lipids are not integral components of the intact NapA dimer. Lipid binding instead occurs via a subset of annular sites that contribute to protein stabilization by varying extents. The stabilizing lipids preferentially pack closely around the dimer domains and the core–dimer interface and these lipids are likely to be held in place by salt bridges to charged residues. In the membrane, these interactions are likely to support the movements of the core domains during transport.

After detergent removal in the gas phase, the lipid-free protein is left largely unsupported, which destabilizes the native fold. In the NapA–lipid complex, lipids protect the protein structure by preventing the collapse of the flexible domain connections normally supported by the bilayer. Charge interactions with lipid head groups are maintained during collisional activation of the protein and provide local stabilization via multiple H-bonds. This is analogous to the protective effects of metal ions on proteins in the gas phase, where multiply coordinated ions reduce the build-up of destabilizing structural rearrangements that occur during collisional activation[26,27]. Unlike metal ions, however, which have a high propensity to form non-specific adducts during desolvation, the complexes between NapA and phospholipids are formed inside detergent micelles in solution, exhibiting an intermediate degree of selectivity for charged lipids bound to a subset of annular positions[28].

The preferred lipid interaction sites and the positions of the most stabilizing lipids identified here are consistent with the locations of cubic phase lipids in NapA and the positions of phospholipids in the X-ray structure of the bacterial citrate symporter CitS, another dimeric elevator-type secondary-active transporter (Supplementary Fig. 7)[7,29]. In the outward-facing structure of CitS, lipid acyl chains insert into crevices between the fixed dimer scaffold and the mobile domain, while the head groups are fixed through interactions with basic residues. This leads us to speculate that the type of lipid interactions described here could serve as structural support during the conformational shifts in elevator-type proteins in general.

The NhaA dimer is held together by interactions between two antiparallel β-hairpins[16,30]. In contrast, NapA has no β-hairpins, and the dimerization interface is instead made up from an additional helix at the N terminus designated TM-1, which interacts with another helix (TM7) on the neighbouring protomer (Supplementary Fig. 8). Because of these structural differences[16,30,31] the dimerization interface in NhaA is less extensive than that seen in NapA. Consistently, our MS data showed that NhaA dimers were easily dissociated following desolvation, whereas NapA dimers were not. Interestingly, NhaA monomers selectively retain CDL, which indicates that these lipids are simply not binding via annular interactions, but are instead incorporated into the protein at specific sites. NapA on the other hand appears lipid free, and the lipids that do interact with the dimer bind to several annular sites.

For eukaryotic $Na^+/H^+$ antiporters, current mechanistic models use the NhaA monomer as a template, since no high-resolution structures exist[32,33]. However, NapA shares higher sequence similarity to human NHA2 than NhaA. Indeed, a sequence alignment and topology prediction of the three antiporters shows that NHA2, like NapA, has additional N-terminal residues that could form a helix in the same position as NapA (Supplementary Fig. 8). Consistently, the MS data indicates that human NHA2 is similar to NapA, since both remain dimeric even at high collision energies and the monomers do not appear to retain any specific lipids, which contrasts the low dimer stability and selective lipid binding displayed by NhaA. Considering the crystallographic evidence for lipid interactions in other elevator proteins (Supplementary Fig. 7), we speculate that the similarities between NapA and NHA2 extend to the role of lipids as structural support between the flexible domain contacts, comparable to the use of membrane lipids as rotary 'bushings' in ATPases[34]. In some antiporters such as NhaA, more specific annular contacts may additionally contribute to their biological functions in the membrane[35].

Taken together, by using MS to inform MD simulations we were able to dissect the relationship between structural features and lipid interactions in the $Na^+/H^+$ exchanger NapA from *T. thermophilus*, which appears to be more similar to the human exchanger NHA2, but not *E. coli* NhaA. Our findings support the hypothesis that eukaryotic $Na^+/H^+$ antiporters share the overall architecture of elevator-type proteins such as NapA, which is of high importance for functional models of these medically important exchangers[1].

## Methods

**Protein preparation.** Cloning and purification of wild-type NapA (MVV), V71C/L141C and I55C NapA, as well as A109T, Q277G, L296M NhaA were performed as described[6,7,36]. Human NHA2 70-480 was cloned into the GAL1-inducible GFP-His8 2 μ vector pDDGFP2 with tobacco etch virus (TEV) cleavage site, transformed into *Saccharomyces cerevisiae* FGY217 strain (MATa, ura3–52, lys2Δ201, pep4Δ) and expressed as described[37]. Twelve litres cultures were grown at 30 °C in URA media with 0.1% glucose. Expression was induced by adding 2% D-galactose, when an $OD_{600}$ of 0.6 was reached. Cells were collected after 22 h, resuspended and lysed by mechanical disruption in 50 mM Tris-HCl pH 7.6, 1 mM EDTA, 0.6 M sorbitol. Membranes were isolated by ultracentrifugation at 200,000*g* for 2 h, homogenized in 20 mM Tris-HCl pH 7.5, 0.3 M sucrose, 0.1 mM $CaCl_2$ and frozen in liquid nitrogen. Thawed

membranes were solubilized by stirring for 2 h at 4 °C in 20 mM Tris, pH 8.0, supplemented with 300 mM NaCl, 100 mM LiCl, 10% glycerol, 2% DDM, and 0.2% cholesteryl hemisuccinate tris salt (CHS). Debris was removed by 45 min centrifugation at 200,000g. A measure of 10 mM imidazole were added to the supernatant and mixed with 1 ml of Ni-NTA Superflow resin (Qiagen) per 1 mg of GFP-His8. The slurry was incubated for 3 h at 4 °C and subsequently loaded washed with 30 column volumes total of 20 mM Tris, pH 8.0 buffer containing 300 mM NaCl, 100 mM LiCl, 1% glycerol, 0.1% DDM, 0.02% CHS and 20 to 30 mM imidazole on a glass Econo-Column (Bio-Rad). The protein was eluted with 250 mM imidazole in 20 mM Tris, pH 8.0 with 300 mM NaCl, 100 mM LiCl, 0.1% glycerol, 0.1% DDM and 0.02% CHS. The eluate was dialysed for 12 h in 3 l 20 mM Tris-HCl buffer, pH 8.0, containing 300 mM NaCl, 100 mM LiCl, 0.03% DDM and 0.006% CHS. The His8 tag was removed during dialysis by adding stoichiometric amounts of His6-tagged TEV protease. After dialysis, TEV protease and the cleaved 8His-tag were removed by passage over a 5 ml Ni-NTA His-Trap column (GE Healthcare). Cleaved NHA2 was concentrated 100 kDa cutoff in spin filters purified by a final size-exclusion step on a Superdex 200 10/300 gel filtration column (GE Healthcare) equilibrated in 20 mM Tris-HCl, pH 8.0, 300 mM NaCl, 100 mM LiCl, 0.006% CHS and 0.03% DDM.

Notably, all proteins were prepared in $>2 \times$ CMC (0.03%) DDM (Generon). Following the final size-exclusion step, protein samples were concentrated to $\sim 0.8$–5 mg ml$^{-1}$ in 20 mM Tris-HCl, pH 7.5 containing 150 mM NaCl and 0.025% DDM using microcentrifuge concentrators with a 100 kDa cutoff. Samples were flash-frozen in liquid N$_2$ and stored at $-80$ °C. Before MS analysis, NapA samples were subjected to solvent and detergent exchanges into 100 mM ammonium acetate pH 7.5 containing 0.02% DDM or 0.5% C8E4. Detergent exchange was carried out at 4 °C with a Superdex Increase 200 column on an Äkta Purifier FPLC system (GE Healthcare). The protein peak was concentrated to 0.6–0.8 ml$^{-1}$, corresponding to 7–10 μM of the NapA dimer, and immediately subjected to MS analysis. Disulphide bonds were reduced through incubation in ammonium acetate/C8E4 supplemented with 10 mM DTT for 30 min at room temperature.

**Lipid stock preparation.** Phospholipids (Avanti Polar lipids, Inc, AL) were dissolved at $\sim 5$ mg of dry material in 1 ml of CHCl$_3$ in a glass vial. Lipid films were generated by solvent evaporation under a stream of N$_2$ and dried in a vacuum chamber over-night. Dry lipid films were solvated in dH$_2$O by multiple rounds of sonication and vortexing and subsequently lyophilized. The resulting lipid cakes were dissolved in 1 ml dH$_2$O and centrifuged for 1 h at maximum speed in a benchtop centrifuge. The supernatant was subjected to phosphate analysis to determine the lipid content, revealing a final lipid concentration between 100 and 500 μM, and stored in 50 μl aliquots at $-20$ °C.

**Mass spectrometry.** Samples were introduced into the mass spectrometer using gold-coated borosilicate capillaries produced in-house. Mass spectra were recorded on hybrid Q-Exactive Orbitrap mass spectrometer (Thermo Fisher) modified for high *m/z* analysis[38,39], or a Synapt G1 T-wave ion mobility mass spectrometer (Waters). The orbitrap settings were: capillary voltage, 1.4 kV; higher energy collisional dissociation (HCD) collision energy 50–200 V as indicated; HCD cell pressure $1 \times 10^{-9}$ mbar; collision gas and argon. Instrument settings were: capillary voltage 1.5 V, cone voltage 130 V, extraction voltage 294 V, collision voltages in the trap raging between 20 and 200 V, and transfer collision voltage 50 V. The source pressure was 5 mbar for native MS and 1.8 mbar for MS/MS. For the MS analysis of DDM-containing protein solutions, the cone voltage was raised to 200 V and the collision voltages ranged between 130 and 200 V. Ion mobility settings were: wave velocity 300 m s$^{-1}$ and wave height 13 V in the ion mobility cell, wave velocity 248 m s$^{-1}$ and wave height 13 V in the transfer region. Drift cell gas was N$_2$ with a pressure of 1.6 torr. CCS calibrations were performed using ADH, Concanavalin A and Pyruvate Kinase (Sigma) and membrane proteins AmtB and pfMATE in C8E4 as calibrants[18,40]. MS data were analysed using Mass Lynx 4.1, DriftScope (Waters, Milford, MA) and PULSAR software packages (http://pulsar.chem.ox.ac.uk/)[41].

**MD simulations.** The alignment of outward-facing NapA crystal structure (PDB code 4BWZ) with respect of the lipid bilayer was determined using OPM server[42]. With this information in hand, the protein was inserted in a $120 \times 95$ Å POPE bilayer using VMD[43]. All lipids having at least one atom within 0.5 Å from the protein (heavy clash) were removed. On the basis of local environment, all histidines were δ protonated. The system was finally solvated and neutralized with 0.15 M NaCl. The resulting system was simulated using CHARMM27 force field using NAMD 2.10 (refs 44,45). Particle-Mesh-Ewald was adopted to treat periodic boundary conditions, and a 12 Å cutoff was applied to non-bonded interactions. The system was first minimized for 500 steepest descent steps. All protein and lipid head atoms were then constrained and a 50 ps *NVT* simulation (300 K, Langevin dynamics, 1 ps$^{-1}$ damping coefficient) was run to melt lipid tails, using 1 fs time step. Subsequently, 250 ps *NPT* simulation (300 K and 1 atm using Langevin piston Nosé-Hoover method, 50 fs piston decay, 200 fs piston period) was run with only protein alpha carbons restrained, again using 1 fs time step. Using the same set-up, all constraint was then

released, and the time step increased to 2 fs for the following 500 ps, using SHAKE to restrain all bonds. Finally, a 150 ns *NPT* production run (300 K, 1 atm) was executed, imposing constant area on the xy plane. Frames extracted every 0.1 ns from the resulting simulation were wrapped and aligned using the protein as reference. The first 50 ns were excluded from analysis, to allow for the lipid bilayer to fully equilibrate. To analyse lipid distribution around the protein, lipids head groups in contact with the protein (at least one atom within 2.15 Å from the protein) at every simulation frame were identified, and the coordinates of their phosphorous atom extracted. Atoms density was determined computing a three-dimensional Gaussian kernel density on all the coordinates using Python. The resulting density was saved in DX format and rendered with a density cutoff of $2 \times 10^{-5}$ nm$^{-3}$. Protein structures were rendered with Chimera[46]. NHA2 topology was predicted using the TMHMM server v 2.0 (http://www.cbs.dtu.dk/services/TMHMM/)[47].

For the gas-phase simulations, the crystal structure of outward-facing NapA (PDB code 4BWZ) was used as a starting structure for subsequent simulation with Gromacs[48]. The simulations, like most classical MD, do not allow for proton transfer. However, no noticeable differences were detected between simulations with different charge states. For moderate deviations from the starting structure, it is therefore assumed that proton transfer is of lesser importance and that classical MD can shed light on the early stages of unfolding. For the lipid-bound system, all lipids with at least one atom within 1 Å distance from the protein were filtered out from the last frame of the MD trajectories of NapA in a PE bilayer. From these, the first annular shell of 23 lipids (10 intracellular and 13 extracellular), in which all lipids make full contact with the protein, were selected by individual inspection. The protonation state of side chains were automatically generated with the pdb2gmx tool, with the exception of E5, H6, H51, E54, E61, E108, E112 and E350 that were protonated to make a neutral system, and additionally E352 and 351 for a $+4$ charge state. The CHARMM27 force field was used to model interatomic interactions, and no cutoffs were used for non-bonded interactions. The structures were energy first minimized with steepest descent, then relaxed in 100 ps simulations at temperatures 300 K, 400 K, (…), 1,000 K with position restraints applied to all non-hydrogen atoms, and subsequently simulated for 10 ns without restraints at the same respective temperatures. The Berendsen thermostat[49] was used for temperature coupling. Bond lengths were constrained with LINCS, and virtual sites were constructed for hydrogen atoms[50], allowing a 3 fs time step. CCS were calculated for the last half of each trajectory using IMPACT, including the trajectory method scaling factor[51].

**Data availability.** The data sets generated and analysed during the current study are available from the corresponding authors on reasonable request.

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

## Acknowledgements
M.L. holds a Marie Curie Early Career Development grant and a Junior Research Fellowship from St Cross College, University of Oxford. E.G.M. holds a Fulford Junior Research Fellowship from Somerville College, University of Oxford, and a Marie Skłodowska Curie INCA Fellowship (grant nr. 2015-00559). M.T.D. is supported by the Swiss National Science Foundation (grant Nr. P2ELP3_155339). J.L.P.B. is a Royal Society University Research Fellow. J.G. is a Junior Research Fellow of The Queen's College and K.G. is a Junior Research Fellow of St Catherine's College, University of Oxford. D.D. acknowledges the support from the EMBO Young Investigator Program, the Swedish Research council, and The Knut and Alice Wallenberg foundation. C.V.R. is supported by a Wellcome Trust Investigator Award (104633/Z/14/Z), an ERC Advanced Grant ENABLE (641317) and an MRC programme grant (MR/N020413/1). We acknowledge the use of the University of Oxford Advanced Research Computing (ARC) facility for this work (http://dx.doi.org/10.5281/zenodo.22558).

## Author contributions
M.L., D.D. and C.V.R. designed the study. M.L., J.G., I.L. and K.G. performed mass spectrometry experiments. E.G.M. and M.T.D. performed molecular dynamics simulations. P.U. and M.C. expressed and purified all proteins. M.L., J.L.P.B., D.D. and C.V.R. analysed the data. M.L., D.D. and C.V.R. wrote the paper with input from all authors.

## Additional information

**Competing financial interests:** The authors declare no competing financial interests.

