## [Peer Review File · Nature Communications]

PEER REVIEW FILE

Reviewers' comments:

Reviewer #1 (Remarks to the Author):

This manuscript presents native mass spectrometry, ion mobility, and molecular dynamics simulation related to the role of lipids in the structure and function of two important Na⁺/H⁺ antiporter membrane proteins. The authors identify both similarities (lipids act as “grease” between domain contacts for both) and differences (NapA is less selective for particular lipid types than is NHA2) between the studied proteins and posit that human NHA2 is more similar to archaeal NapA—and less similar to *E. coli* NhaA—than previously believed. From gas phase experiments, the authors infer a strong dimer interface for NHA2 and NapA and a weaker one for NhaA, and NhaA appears to bind cardiolipin strongly while the other two complexes appear to have weaker lipid binding. These are undoubtedly very interesting conclusions for a broad readership, and there are many positive aspects of the manuscript.

Pushing the limits of native mass spectrometry and ion mobility, simulating membrane protein-lipid interactions, and simulating gas-phase dissociation of complicated assemblies are all exciting endeavors, and this manuscript does well in showing how physiological questions can be addressed with these methods. The data and methodology are sound. All of the figures are very clear, and the references are appropriate. However, I have concerns about the strength of the evidence for the numerous “speculations” in the manuscript and suggest the authors provide stronger evidence for them or eliminate them altogether.

A few major concerns I have with the conclusions of the manuscript that should be addressed before publication in Nature Communications are:

- 1) What evidence is there that collisional dissociation of the detergent-lipid micelle from the complex does not scramble the locations of the lipids away from what they would be natively in solution? Especially for charged lipids (PG), it seems at least highly plausible that lipids could migrate to form strong salt-bridge-like interactions with positively charged residues on the protein that may be far from their initial locations, perhaps even far from the transmembrane region.

(Although not the same, something analogous to this is believed to occur in “proton hopping” models for collisional dissociation of proteins.) In this case, these gas-phase findings may not be related to physiological protein-lipid interactions. Results from, e.g., Russell and coworkers (doi: 10.1021/acs.analchem.6b00871) illustrate that gas-phase salt-bridge-like interactions between proteins and charged adducts can lead to high gas-phase conformational stability that may have little to do with solution behavior. One could imagine this leading to especially strong artifacts for cardiolipin, which may have two (covalently linked) salt-bridge forming sites—if both sites are strongly bound in the gas phase, with or without scrambling or migration, this lipid may be especially difficult to dissociate without any clear physiological relevance.

2) If lipids can be packed between the dimer and core domains (p. 9), is their resistance to dissociation evidence of a particularly strong binding or of a kinetic preference for more exposed lipids to move and/or dissociate?

3) The vacuum MD simulations do not appear to allow for migration of protons during the heating process, though this process is strongly implicated in gas-phase collisional unfolding and dissociation. How are salt-bridge interactions between negatively charged lipids and positive residues treated? Would they be able to dissociate as neutrals by proton transfer in these simulations? It would seem important for them to be able to do so for reliable and accurate modeling of the lipid-protein interactions and dissociation.

4) On p. 13, the authors claim that their data suggest a 13-helix architecture for NHA2 rather than the previously assumed 12-helix architecture based on NhaA. It is not clear to me to what part of their data they are referring here. Is this claim based on the generally similar gas-phase behavior of NHA2 and NapA (which is clearly not resolved to the level of secondary or detailed tertiary structure in these experiments), or based on the homology modeling alone? The support for this claim is either largely absent or unclear.

5) Similarly, the argument about matching helical pitch for cardiolipin acyl chains and NhaA sidechains (p. 13) seems very weakly supported. Although it is plausible, there does not appear to be especially strong evidence here for or against a claim at this level of structural detail. By contrast, the speculation about the lipids as “molecular grease” is more appropriate for the level of detail in the data presented in the manuscript.

I think the manuscript would benefit from separating much more clearly the strongly supported conclusions from the weaker or entirely speculative ones, perhaps by moving all speculation to a single paragraph or two at the end of the manuscript rather than intermixing it with firmer conclusions. This would also more clearly establish the current capabilities and limitations of

these sorts of experiments and simulations.

Reviewer #2 (Remarks to the Author):

The manuscript by Landreh et al describes a study of lipid protein interaction in three transporters, NhaA, NapA, and NHA2, all of which are thought to utilize an elevator type mechanism to transport ions. While the main method is ion mobility mass spectrometry (IM-MS), molecular dynamics simulations were applied to either extract new information or to validate and fortify conclusions from the IM-MS. The authors first showed that the IM-MS can detect and resolve lipids associated with the transport proteins, and that in two of the three transporters dimeric assembly can be preserved under relatively mild ionization conditions. The authors then went on to interpret an increase of arrival time to an increase of collision cross section and in turn to unfolding of the proteins, and showed that the presence of lipids can reduce the unfolding. The authors then showed that a negatively charged lipid is preferred and additional lipids further stabilize the folded state. MD simulations seem to indicate that the negatively charged lipid tend to stay at the interface between the two structurally distinct domains. The main conclusion of the study is that negatively charge lipids bind at the interface of the two domains to either provide structural support or to facilitate the relative motions between the two domains. Overall, the data that indicate clearly resolved protein-bound lipids are strong, and the conclusions are reasonable. However, the connection between the conclusions and observations are not rigorous. The major issue is that there is a significant compaction of the protein in the vacuum and it is not clear how the unfolding of the compacted protein differ from these in the membrane, and further, whether the unfolding in vacuum is related in anyway to the relative movement of the two domains. On top of that, the crosslinked protein itself can unfold, presumably taking a different pathway, in similar mass spec conditions, which would further complicates data interpretation.

Reviewer #3 (Remarks to the Author):

The manuscript by Landreh et al provides a detailed analysis of the mass spectral properties of three sodium-proton antiporter proteins, E. coli NhaA, thermophilic NapA and mammalian NHA2. Structures are available for the former two proteins, but not for the latter, biomedically important protein. Landreh et al provide molecular interpretation for their MS data by the use of molecular dynamics simulations of one of the structures of NapA, both in a POPE lipid bilayer and in vacuum. Overall, the data is novel, well-presented, well-described with appropriate citations.

The MS data seems to be very insightful, in particular in terms of the monomer/dimer equilibrium of the three transporters, and the preference for specific-high affinity lipid interactions of those complexes. Specifically, they show that NhaA dimers are unstable, but that lipids interact strongly with the monomers, interestingly, cardiolipin molecules appear to be a major player. For NapA and NHA2, although the signal for the latter is more noisy, there is clear evidence for stable dimers, but with few lipid adducts remaining in place. Given the large “elevator-like” conformational changes identified crystallographically for NapA, the stability of the dimer would appear to be of critical importance for aiding in that process.

The second major component of the manuscript is an analysis of the “unfolding” of NapA as the collision voltage is increased, comparing the wild type protein with different conditions, such as disulphide cross-linking and the presence of added lipids of different head group compositions. These experiments are well-designed although there are some odd effects on the unfolding profile of adding DTT on the disulphide-containing protein. The authors should comment on this, and on the absence of a similar experiment using DTT on the other cross-linked double-cysteine mutant - I presume they tried it and are not showing it for some reason.

My major concern is the description of involvement of the lipids in stabilizing the protein. This comes in two stages, based analysis of the MD simulations with a PE bilayer, and then based on the in vacuo simulations.

Regarding the data relating to the PE-bilayer simulations:

- I note, for future studies, that Charmm36 would be a preferable choice of force field for lipids, especially when considering packing and compression, although I can accept that the use of Charmm27 is not prohibitive to publication.

However, the following points should be addressed:

- First, as stated in the methods, the authors present the phosphate distribution/density, not the lipid distribution, and so the text should be rewritten accordingly. It is entirely expected that the phosphates collect near the positive charges in the protein - this is what the force field will always do with such charge-charge distributions, especially when all the lipids are negatively charged. Therefore it is not really fair to describe this as “good agreement” with the importance of head-group charge that was observed by MS (line 222). If the simulation system contained a mixture of uncharged/zwitterionic/anionic head groups and then there was still an increased density for the anionic head groups near those groups, that would be stronger evidence. This should be rephrased.

- Second, there cannot be a vacuum, so are the remaining spaces filled with choline or with lipid tails or water?

- Third, I found figure 4a difficult to interpret - are some of the densities are on one side of the protein/membrane only? That is, next to the loop being described? Perhaps those could be colored differently according to the leaflet that they come from to make this clearer.

- Fourth, the authors interpret this apparent increase in phosphate density (and the compression of the lipids near the dimer domain) as NapA preferentially interacting with lipids that "tightly pack" into the connection between the dimer and core domains. To me this is an over-interpretation. First, the densities in Figure 4a are smeared over quite a wide range of the protein surface (including the transport domain, on the left of the figure), and not focused exclusively on the dimer/core domain. Moreover, compression doesn't mean that the interaction with the protein is tight, simply that the lipid head groups have to be closer to each other on the two sites because of the hydrophobic mismatch. I think the authors need to add more data if they want to demonstrate such an increase in affinity/interaction. Alternatively, they should tone down these conclusions.

Regarding the interpretation of the in vacuo data/simulations:

- I think it should be made more clear that the protein does behave differently in vacuum (it collapses), and therefore there is some uncertainty as to whether the interactions inferred from its MS unfolding behavior are an artifact of it being in vacuo. Specifically, the question that needs to be more clearly addressed is: what is the relevance of the in vacuo lipid effects to the native behavior/mechanism of the protein (which is also reported to be "lipid-free").

- Relatedly, and importantly, there seems to me to be some discrepancy/fuzziness between the abstract/conclusions and the description of the data. For example, a single (or two) PE molecule changes the "unfolding" behavior of NapA significantly (Fig. 3e), and in Figure 4, the authors state that lipids "prevent structural collapse ... by intercalating into the interface between core and dimer domains" and "mediate contacts between the mobile protein segments" (line 37), and yet, in other places they appear to be saying that there is no evidence for specific lipids, and that the effects reflect multiple interactions (e.g. analyzed in terms of hydrogen-bonds per head group in the 23 lipids), e.g. from a "subset of annular sites that contribute to varying extents" (line 278). The writing should be made more consistent throughout to clarify how far these conclusions can really go, or to what extent some of these results may be artifactual.

- I find the term "molecular grease" confusing, and the evidence for it unclear. Most of the quantitative analysis relates to strong salt bridge and hydrogen-bond interactions, rather than the hydrophobic tails forming simultaneous interactions with both domains, which is illustrated by a snapshot (Fig. 6c) and by analogy with dehydrated crystal structures (Supplement). Thus, it seems a stretch to say that the lipids are acting as grease; if they are stabilizing the flexible linkers, then they are acting more like anchors, not grease.

In terms of the set up of the in vacuo calculations themselves:

- First, the authors describe the choice of 23 lipids as "representing the entire range of preferential positions observed in MD" (line 229). Some supporting evidence should be provided for that statement.

- Second, it is written that the lipids have "sufficient space to rearrange on the surface of the protein" (line 230). I don't understand what "rearrangement" means in this context, and why that

is required/desirable. Is the assumption that the collisions during the Mass Spec would cause such rearrangements, or that going into vacuum might require such rearrangements, or something else? In either case, are the authors saying that a 10ns/20ns simulation would capture those differences reliably? The stability of the data with respect to temperature (Fig. 4e) would seem to support that.

- I assume that the error bars shown in Figure 4e relate to the collisional cross section (CCS) values obtained during the 10ns simulation, not due to multiple trajectories; either way, this should be stated in the legend.

Minor points:

- I found the use of the term “apo” confusing. On first usage, it should be highlighted that the authors mean lipid-free, as opposed to sodium or protons.

- Similarly, the legend of figure 1 should explain that the red spheres reflect lipids.

Reviewers' comments:

Reviewer #1 (Remarks to the Author):

1) What evidence is there that collisional dissociation of the detergent-lipid micelle from the complex does not scramble the locations of the lipids away from what they would be natively in solution? Especially for charged lipids (PG), it seems at least highly plausible that lipids could migrate to form strong salt-bridge-like interactions with positively charged residues on the protein that may be far from their initial locations, perhaps even far from the transmembrane region. (Although not the same, something analogous to this is believed to occur in “proton hopping” models for collisional dissociation of proteins.) In this case, these gas-phase findings may not be related to physiological protein-lipid interactions. Results from, e.g., Russell and coworkers (doi: 10.1021/acs.analchem.6b00871) illustrate that gas-phase salt-bridge-like interactions between proteins and charged adducts can lead to high gas-phase conformational stability that may have little to do with solution behavior. One could imagine this leading to especially strong artifacts for cardiolipin, which may have two (covalently linked) salt-bridge forming sites—if both sites are strongly bound in the gas phase, with or without scrambling or migration, this lipid may be especially difficult to dissociate without any clear physiological relevance.

The reviewer raises an interesting point. It is widely accepted that charge-based interactions, such as salt bridges, are important factors in maintaining the protein fold in the gas phase. Consequently, additional salt bridges, either with low-MW ligands or adduct ions, can help stabilize proteins after desolvation. We state in the text that the stabilizing effect of lipids is analogous to the effects of multivalent ions on soluble proteins (p. 10). We now add a citation to the study of anion binding by Russell et al; this is in addition to the paper by Ruotolo et al describing cation effects (ref. 25 and 26 in the revised manuscript). The protective effects of adduct ions in the above studies stems from non-specific interactions as the ions are deposited on charged residues at the surface of the protein during the last stages of the electrospray process. This process is distinct from the observations we make for lipid binding.

[redacted]

For NapA virtually all of the surface-exposed basic residues are located at the border of the transmembrane region and near the domain interfaces. The MD simulations in this study show that this particular subset of annular positions is the most populated in solution, The phosphate head-groups of the lipids bound to these sites will therefore be highly coordinated by the neighbouring basic residues. Given that there is a positive correlation between ligand coordination and gas phase stabilization, we reason that the positions

preferred in solution are likely the positions that provide the strongest gas phase stabilization.

In our gas-phase MD simulations, we observe that while the acyl chains migrate and finally detach from the protein, lipid head-groups remain attached to their initial basic sites during collisional unfolding. During later stages of unfolding (here represented by temperatures above 600 K), we find that further neighbouring charged residues orient themselves towards the bound lipid head-groups, effectively increasing the local stabilization (see Figure 4 f). In summary, we believe that the observed effects of lipids on the structure of NapA can be rationalized with the current thinking about interactions in the gas phase. By extrapolating these concepts into the lipid environment we can probe the roles of lipids in the three different transporters described here.

2) If lipids can be packed between the dimer and core domains (p. 9), is their resistance to dissociation evidence of a particularly strong binding or of a kinetic preference for more exposed lipids to move and/or dissociate?

Again this is a good point and one that we also considered. We reason that the preferential binding of lipids to the regions around the core-dimer interface stems from the higher number of energetically favourable interactions at these sites, i.e. between the phosphate head-groups and basic side-chains. There is also the possibility to pack acyl chains into the crevice between the dimer and core domains. Lipids that form such tight interactions with the protein are more likely to be retained during detergent removal inside the mass spectrometer. We therefore conclude that the lipids that have lower propensity to dissociate in the gas phase are attached via the particularly favourable interactions that are responsible for preferential protein-lipid contacts in solution.

3) The vacuum MD simulations do not appear to allow for migration of protons during the heating process, though this process is strongly implicated in gas-phase collisional unfolding and dissociation. How are salt-bridge interactions between negatively charged lipids and positive residues treated? Would they be able to dissociate as neutrals by proton transfer in these simulations? It would seem important for them to be able to do so for reliable and accurate modelling of the lipid-protein interactions and dissociation.

Indeed, the MD simulations do not allow for migration of protons, and consequently the gas-phase unfolding seen in the simulations may in some cases deviate from what takes place in experiments. It should be noted however that we have conducted the gas-phase MD experiments with two different protonation states and obtained identical outcomes. There have been a few reports describing MD of macromolecules with proton transfer but such techniques are still in their infancy. Given the challenges in creating such tools, we are not in a position to resolve this issue specifically for this manuscript. The reviewer's points are fair and important, and we have revised the results section to discuss this issue. See p. 10.

4) On p. 13, the authors claim that their data suggest a 13-helix architecture for NHA2 rather than the previously assumed 12-helix architecture based on NhaA. It is not clear to me to what part of their data they are referring here. Is this claim based on the generally similar gas-phase behavior of NHA2 and NapA (which is clearly not resolved to the level of secondary or detailed tertiary structure in these experiments), or based on the homology modeling alone? The support for this claim is either largely absent or unclear.

This is a valid point. Although we believe that NHA2 shares the 13-helix architecture of NapA, we should in hindsight not have suggested such a detailed structural model. We have therefore carefully revised this section to clarify the origin of our simplified model (p. 14). Despite the overall similarity of all three proteins, both NapA and NHA2 contain an N-terminal extension not present in NhaA (see ref. 6). In NapA, this extension forms an additional helix (numbered -1) that comprises a large part of the dimer interface in NapA. The additional protomer interactions provided by helix -1 result in the stable dimer architecture of NapA that is absent in NhaA. Based on the sequence alignment of the N-terminal regions of NhaA, NapA, and NHA2 (new Figure S8), as well as TMHMM analysis, which predicts a transmembrane region in the corresponding part of NHA2, we propose that NHA2 also contains an additional N-terminal helix that gives rise to a similarly stable dimer as in NapA.

5) Similarly, the argument about matching helical pitch for cardiolipin acyl chains and NhaA sidechains (p. 13) seems very weakly supported. Although it is plausible, there does not appear to be especially strong evidence here for or against a claim at this level of structural detail. By contrast, the speculation about the lipids as “molecular grease” is more appropriate for the level of detail in the data presented in the manuscript.

We have considered the reviewer’s suggestion and agree that discussing the relationship between acyl chain length and helical pitch, although intriguing in the light of our observations, may raise further issues that distract from the main focus of the manuscript. We have removed the corresponding section from p. 14.

I think the manuscript would benefit from separating much more clearly the strongly supported conclusions from the weaker or entirely speculative ones, perhaps by moving all speculation to a single paragraph or two at the end of the manuscript rather than intermixing it with firmer conclusions. This would also more clearly establish the current capabilities and limitations of these sorts of experiments and simulations.

We have followed the advice from reviewers 1 and 3 and re-structured the discussion on p. 13 and 14 to distinguish the experiment-derived conclusions from the more speculative points.

Reviewer #2 (Remarks to the Author):

Overall, the data that indicate clearly resolved protein-bound lipids are strong, and the conclusions are reasonable. However, the connection between the conclusions and observations are not rigorous. The major issue is that there is a significant compaction of the protein in the vacuum and it is not clear how the unfolding of the compacted protein differ from these in the membrane, and further, whether the unfolding in vacuum is related in anyway to the relative movement of the two domains. On top of that, the crosslinked protein itself can unfold, presumably taking a different pathway, in similar mass spec conditions, which would further complicates data interpretation.

The reviewer raises an important aspect of gas phase structural biology. The compaction of proteins in the absence of solvent is well-established and can be attributed to the collapse of charged side-chains and unsupported loops (see e.g. Breuker & McLafferty 2008, doi:10.1073/pnas.0807005105). This is faithfully captured by our MD simulations of the lipid-free protein in vacuo. The observed 10-15% reduction in CCS is anticipated for a folded protein (see ref. 41). As much as >30% compaction has also been reported for loosely folded proteins (see Barran et al 2016, doi:10.1038/ncomms12163). Taking the gas-phase compaction into account, we are able to investigate the effects of lipids on the gas phase structure of NapA and relate our findings to the situation in the membrane (see our response to reviewer 1).

Reviewer #3 (Remarks to the Author):

The second major component of the manuscript is an analysis of the “unfolding” of NapA as the collision voltage is increased, comparing the wild type protein with different conditions, such as disulphide cross-linking and the presence of added lipids of different head group compositions. These experiments are well-designed although there are some odd effects on the unfolding profile of adding DTT on the disulphide-containing protein. The authors should comment on this, and on the absence of a similar experiment using DTT on the other cross-linked double-cysteine mutant - I presume they tried it and are not showing it for some reason.

This is an interesting point. The I55C mutant was chosen to investigate the effects of disulfide reduction as it carries a single disulfide bond per dimer in a solvent-accessible region, as opposed to two bonds per dimer in the hydrophobic interface between the core and transport domains in the V71C/L141C mutant. However, as the reviewer points out, the unfolding transitions of the reduced I55C mutant and the arrival time distribution of the unfolded species are broader than for the wt. This suggests that even for the I55C mutant, we achieve only incomplete reduction of the disulfide bond at high DTT concentrations as observed by SDS PAGE (see Figure S1 in reference 7 Coincon et al 2016, doi: 10.1038/nsmb.3164). We therefore agree with the reviewer that the unfolding profile of the I55C mutant in the presence of DTT, although very similar, is not truly identical to that of the wild-type protein. We now discuss this issue in detail on p. 7.

As stated in the methods, the authors present the phosphate distribution/density, not the lipid distribution, and so the text should be rewritten accordingly. It is entirely expected that the phosphates collect near the positive charges in the protein - this is what the force field will always do with such charge-charge distributions, especially when all the lipids are negatively charged. Therefore it is not really fair to describe this as “good agreement” with the importance of head-group charge that was observed by MS (line 222). If the simulation system contained a mixture of uncharged/zwitterionic/anionic head groups and then there was still an increased density for the anionic head groups near those groups that would be stronger evidence. This should be rephrased.

The reviewer makes a valid point. The wording on pp. 9-10 was changed accordingly to clarify that we do not present the fact that the negatively charged head-groups cluster around basic residues as a surprising finding, but rather a case-in-point that the basic residues mark the preferred lipid binding sites (see also reference 26).

Second, there cannot be a vacuum, so are the remaining spaces filled with choline or with lipid tails or water?

The structures are shown as cylinder or ribbon renderings based on the backbone only. Although it may appear in these images as if the core and dimer domains are separated by a large cavity, they are actually in direct contact with each other. The empty spaces are filled with bulky hydrophobic side-chains. We interpret the fact that we see lipid insertion into this tight interface as part of the “molecular grease” function.

Third, I found figure 4a difficult to interpret - are some of the densities are on one side of the protein/membrane only? That is, next to the loop being described? Perhaps those could be colored differently according to the leaflet that they come from to make this clearer.

The high lipid densities are in fact observed next to the loop that functions as the hinge between core and transport domain. We have followed the reviewer’s suggestion and colored the densities on the intra- and extracellular sides differently.

Fourth, the authors interpret this apparent increase in phosphate density (and the compression of the lipids near the dimer domain) as NapA preferentially interacting with lipids that “tightly pack” into the connection between the dimer and core domains. To me this is an over-interpretation. First, the densities in Figure 4a are smeared over quite a wide range of the protein surface (including the transport domain, on the left of the figure), and not focused exclusively on the dimer/core domain. Moreover, compression doesn’t mean that the interaction with the protein is tight, simply that the lipid head groups have to be closer to each

other on the two sites because of the hydrophobic mismatch. I think the authors need to add more data if they want to demonstrate such an increase in affinity/interaction. Alternatively, they should tone down these conclusions.

We understand the reviewer's concern. Our simulations suggest that the dimer/core domain area is the most consistently populated, but understand that the observed densities represent a fluctuating continuum. Regarding the compression, we agree that compression means reduced membrane thickness, but highlight the finding that the most consistently populated sites coincide with the highest degree of hydrophobic mismatch. These points are now clarified (p. 9 and in the discussion).

I think it should be made clearer that the protein does behave differently in vacuum (it collapses), and therefore there is some uncertainty as to whether the interactions inferred from its MS unfolding behavior are an artifact of it being in vacuo. Specifically, the question that needs to be more clearly addressed is: what is the relevance of the in vacuo lipid effects to the native behavior/mechanism of the protein (which is also reported to be “lipid-free”).

As outlined in responses to reviewers 1 and 2, the compaction of the protein in vacuum is in good agreement with the proposed structural re-arrangements that occur during desolvation, attributed to the collapse of charged side-chains and unsupported loop regions. Lipids that are bound by NapA in solution are retained and likely become more tightly incorporated as the protein undergoes compaction after desolvation. The MD simulations in this study show how the lipid interactions that occur in solution can give rise to lipid-mediated stabilization of the protein structure. The gas-phase effects have been taken fully into account (see also our response to reviewer 1). The fact that the protein appears “lipid-free” in ESI-MS and when crystallized from detergent suggests that it does not incorporate structural lipids ie those that are critically required to maintain an intact complex. Instead, exogenous lipids help to stabilize the dynamic regions of the protein. It should be noted that this appears to be a key difference between NapA and NhaA (only the latter is extracted with CDL present) underscoring the validity of the approach.

Relatedly, and importantly, there seems to me to be some discrepancy/fuzziness between the abstract/conclusions and the description of the data. For example, a single (or two) PE molecule changes the “unfolding” behavior of NapA significantly (Fig. 3e), and in Figure 4, the authors state that lipids “prevent structural collapse ... by intercalating into the interface between core and dimer domains” and “mediate contacts between the mobile protein segments” (line 37), and yet, in other places they appear to be saying that there is no evidence for specific lipids, and that the effects reflect multiple interactions (e.g. analyzed in terms of hydrogen-bonds per head group in the 23 lipids), e.g. from a “subset of annular sites that contribute to varying extents” (line 278). The writing should be made more consistent throughout to clarify how far these conclusions can really go, or to what extent some of these results may be artifactual.

We observe that bound lipids both reduce the structural collapse of NapA, after transfer to the gas phase, and stabilize the protein against unfolding during collisional activation. Regarding the lack of evidence for specific lipid interactions, we find that lipids in a subset of annular positions including the domain interfaces can exert these protective effects (i.e. neither a single type of lipid nor via a single high-affinity site). This two-fold effect of lipids on the desolvated protein is now clarified pp. 11-13).

I find the term “molecular grease” confusing, and the evidence for it unclear. Most of the quantitative analysis relates to strong salt bridge and hydrogen-bond interactions, rather than the hydrophobic tails forming simultaneous interactions with both domains, which is illustrated by a snapshot (Fig. 6c) and by analogy with dehydrated crystal structures (Supplement). Thus, it seems a stretch to say that the lipids are acting as grease; if they are stabilizing the flexible linkers, then they are acting more like anchors, not grease.

We have changed the wording on pp. 14-15 and now describe lipids in antiporters more generally as providers of “structural support” to avoid confusion.

The authors describe the choice of 23 lipids as “representing the entire range of preferential positions observed in MD” (line 229). Some supporting evidence should be provided for that statement.

To choose the lipids for gas-phase MD, we used the last frame from the 150 ns all-atom simulation of NapA in a PE bilayer and selected all lipids with at least one atom within 1 Å of the protein. We then manually selected the lipids from both leaflets around the transmembrane domain that are in full contact with the protein, i.e. the first annular shell. We have clarified this in the text and the method section now.

It is written that the lipids have “sufficient space to rearrange on the surface of the protein” (line 230). I don’t understand what “rearrangement” means in this context, and why that is required/desirable. Is the assumption that the collisions during the Mass Spec would cause such rearrangements, or that going into vacuum might require such rearrangements, or something else? In either case, are the authors saying that a 10ns/20ns simulation would capture those differences reliably? The stability of the data with respect to temperature (Fig. 4e) would support that.

By “rearrange” we mean local repositioning of entire lipids or parts thereof in response to the transfer to the gas phase. We do not consider such rearrangements to be (un)desirable, nor do we assume that they are required, but we acknowledge that they may occur. It is difficult to know what the required simulation time to recapitulate this process is, but as the reviewer suggests the stability with respect to temperature suggests that we capture the

majority of such rearrangement, at least those taking place on moderate time scales. See revisions on pp. 10-11.

I assume that the error bars shown in Figure 4e relate to the collisional cross section (CCS) values obtained during the 10ns simulation, not due to multiple trajectories; either way, this should be stated in the legend.

We apologise for this omission. The error bars reflect variation along each trajectory. We clarify this and also the fact that the first half of each simulation was excluded from analysis. See p. 25.

Minor points:

I found the use of the term “apo” confusing. On first usage, it should be highlighted that the authors mean lipid-free, as opposed to sodium or protons.

Similarly, the legend of figure 1 should explain that the red spheres reflect lipids.

We have made all corresponding changes.

Reviewers' Comments:

Reviewer #1 (Remarks to the Author):

I find that the authors have satisfactorily addressed the concerns raised by the reviewers, and I recommend the revised manuscript for publication in Nature Communications.

Reviewer #2 (Remarks to the Author):

I have no further comments.

Reviewer #3 (Remarks to the Author):

The authors have improved the manuscript significantly in their revisions and clarified most of my questions, but there remain a few areas in the manuscript where I believe rephrasing is needed.

The abstract contains an apparent inconsistency, relating to the point about specific and general effects of lipids (which has been clarified now in the main text). Specifically, the abstract states that NapA forms "stable, lipid-free dimers", but also later, that (1) the dimer interface in NapA is stabilized against unfolding by lipids at the domain interface, and (2) NapA uses annular lipids as a structure support to facilitate large-scale motions. The latter two statements seem to be incompatible with the former observation that the dimers are lipid-free and yet stable. To clarify: do the authors mean that NapA can be stable without lipids under equilibrium conditions (in detergent micelles/in vacuo?), but are stabilized by lipids under unfolding conditions, in ways that suggest that the conformational change would also be stabilized by lipids? Actually, in the main text, the authors mention that the NapA monomers retain only very minor adducts [line 114 - do they mean "minor amounts of lipid adducts"?], which implies that there are some lipids attached to the monomer, and that they are not lipid-free - are these the ones that stabilize the protein at higher collisions? In other words, under the "gentler" equilibrium conditions the protein is not undergoing any conformational changes - is this the right interpretation? Rephrasing should clarify these points, and would make the abstract more in line with the rest of the manuscript.

The introduction also seems inconsistent with the discussion, because they "conclude" rather than speculate that the lipids "provide structural support that facilitates the large-scale conformational changes" (line 83). The word conclude seems too strong relative to how it is

phrased in the discussion, which better agrees with the fact that the data does not relate to the conformational change per se, but rather to unfolding.

Other minor corrections, to improve the clarity of the conclusions and the reproducibility:

- how exactly were the density figures made, e.g. with what cutoffs? To be reproducible, this information should be provided somewhere.

- Line 81-82: the lipids “stabilize the flexible attachment” and “specifically pack around the mobile segments”; what is meant by the mobile segments here? Do the authors mean that they interact both with the mobile core domain AND the flexible attachment, but that they only stabilize the flexible attachment?

- Line 145-148: The consistency between calculated and experimental values support the idea that NapA retains its native conformation, except for the collapse of surface residues and unsupported loops, i.e. do the authors mean that these two are consistent in the sense that the protein is not unfolded?

- typo on line 457: “out all lipids”

Reviewers' comments:

Reviewer #3 (Remarks to the Author):

The authors have improved the manuscript significantly in their revisions and clarified most of my questions, but there remain a few areas in the manuscript where I believe rephrasing is needed.

The abstract contains an apparent inconsistency, relating to the point about specific and general effects of lipids (which has been clarified now in the main text). Specifically, the abstract states that NapA forms "stable, lipid-free dimers", but also later, that (1) the dimer interface in NapA is stabilized against unfolding by lipids at the domain interface, and (2) NapA uses annular lipids as a structure support to facilitate large-scale motions. The latter two statements seem to be incompatible with the former observation that the dimers are lipid-free and yet stable. To clarify: do the authors mean that NapA can be stable without lipids under equilibrium conditions (in detergent micelles/in vacuo?), but are stabilized by lipids under unfolding conditions, in ways that suggest that the conformational change would also be stabilized by lipids? Actually, in the main text, the authors mention that the NapA monomers retain only very minor adducts [line 114 - do they mean "minor amounts of lipid adducts"?], which implies that there are some lipids attached to the monomer, and that they are not lipid-free - are these the ones that stabilize the protein at higher collisions? In other words, under the "gentler" equilibrium conditions the protein is not undergoing any conformational changes - is this the right interpretation? Rephrasing should clarify these points, and would make the abstract more in line with the rest of the manuscript.

We understand the reviewer's concern and agree that the phrase "stable, lipid-free dimers" appears to be at odds with the observation that lipid binding provides conformational stabilization of NapA. The statement refers to two observations: 1.) NapA forms stable dimers that can be preserved in the gas phase, unlike NhaA, which nearly completely dissociates. 2.) NapA retains only very minor amounts of lipid adducts after purification with no apparent specificity, again unlike NhaA, which selectively retains significant amounts of cardiolipin. To resolve the apparent inconsistency, we have rephrased the abstract and now state these two observations separately. The wording in line 114 has been changed to "very minor amounts of lipid adducts".

The introduction also seems inconsistent with the discussion, because they "conclude" rather than speculate that the lipids "provide structural support that facilitates the large-scale conformational changes" (line 83). The word conclude seems too strong relative to how it is phrased in the discussion, which better agrees with the fact that the data does not relate to the conformational change per se, but rather to unfolding.

Again, the reviewer raises a valid observation. We have changed the wording in the abstract to "speculate".

Other minor corrections, to improve the clarity of the conclusions and the reproducibility:

- *how exactly were the density figures made, e.g. with what cutoffs? To be reproducible, this information should be provided somewhere.*

Density cutoffs are now included in the methods section.

- *Line 81-82: the lipids “stabilize the flexible attachment” and “specifically pack around the mobile segments”;* what is meant by the mobile segments here? Do the authors mean that they interact both with the mobile core domain AND the flexible attachment, but that they only stabilize the flexible attachment? Our results suggest that the lipids form contacts with both the core and dimer domain by binding at the flexible hinge region. This is now stated clearly.
- *Line 145-148: The consistency between calculated and experimental values support the idea that NapA retains its native conformation, except for the collapse of surface residues and unsupported loops, i.e. do the authors mean that these two are consistent in the sense that the protein is not unfolded?* Yes, our conclusions are in line with an extensive body of work showing that proteins can retain an overall native-like conformation in the gas phase following gentle ionization (see e.g. references 20-23). The observed moderately reduced CCS compared to the crystal structure is due to the collapse of unsupported loops and side-chains. Significant unfolding events, on the other hand, would result in a larger CCS.
- *typo on line 457: “out all lipids”*
We have corrected the typo by removing the word “out”.